# EasyTPP: Towards Open Benchmarking Temporal Point Processes

**Siqiao Xue**[◇], **Xiaoming Shi**[◇], **Zhixuan Chu**[◇], **Yan Wang**[◇], **Hongyan Hao**[◇], **Fan Zhou**[◇],
**Caigao Jiang**[◇], **Chen Pan**[◇], **James Y. Zhang**[◇], **Qingsong Wen**[♠], **Jun Zhou**[◇]
[◇]Ant Group,  [♠]Alibaba Group
`siqiao.xsq@alibaba-inc.com`

**Hongyuan Mei**
TTIC
`hongyuan@ttic.edu`

## Abstract

Continuous-time event sequences play a vital role in real-world domains such as healthcare, finance, online shopping, social networks, and so on. To model such data, temporal point processes (TPPs) have emerged as the most natural and competitive models, making a significant impact in both academic and application communities. Despite the emergence of many powerful models in recent years, there hasn't been a central benchmark for these models and future research endeavors. This lack of standardization impedes researchers and practitioners from comparing methods and reproducing results, potentially slowing down progress in this field. In this paper, we present EasyTPP, the first central repository of research assets (e.g., data, models, evaluation programs, and documentation) in the area of event sequence modeling. Our EasyTPP makes several unique contributions to this area: a unified interface of using existing datasets and adding new datasets; a wide range of evaluation programs that are easy to use and extend as well as facilitate reproducible research; implementations of popular neural TPPs, together with a rich library of modules by composing which one could quickly build complex models. We will actively maintain this benchmark and welcome contributions from other researchers and practitioners. Our benchmark will help promote reproducible research in this field, thus accelerating research progress as well as making more significant real-world impacts. The code and data are available at `https://github.com/ant-research/EasyTemporalPointProcess`.

## 1 Introduction

Continuous-time event sequences are ubiquitous in various real-world domains, such as neural spike trains in neuroscience (Williams et al., 2020), orders in financial transactions (Jin et al., 2020), and user page viewing behavior in the e-commerce platform (Hernandez et al., 2017). To model these event sequences, temporal point processes (TPPs) are commonly used, which specify the probability of each event type's instantaneous occurrence, also known as the *intensity function*, conditioned on the past event history. Classical TPPs, such as Poisson processes (Daley &

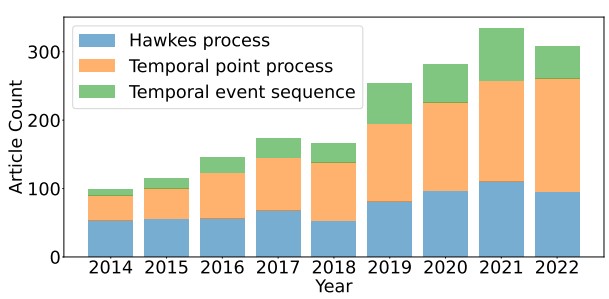

Figure 1: ArXiv submissions over time on TPPs. See Appendix E.1 for details.

Vere-Jones, 2007) and Hawkes processes (Hawkes, 1971), have a well-established mathematical foundation and have been widely used to model traffic (Cramér, 1969), finance (Hasbrouck, 1991) and seismology (Ogata, 1988) for several decades. However, the strong parametric assumptions in these models constrain their ability to capture the complexity of real-world phenomena.

To overcome the limitations of classical TPPs, many researchers have been developing neural versions of TPPs, which leverage the expressiveness of neural networks to learn complex dependencies; see section 7 for a comprehensive discussion. Since then, numerous advancements have been made in this field, as evidenced by the rapidly growing literature on neural TPPs since 2016. Recent reviews have documented the extensive methodological developments in TPPs, which have expanded their applicability to various real-world scenarios. As shown in Figure 1 (see Appendix E.1 for details of how we count the articles.), the number of research papers on TPPs has been steadily increasing, indicating the growing interest and potential impact of this research area. These advancements have enabled more accurate and flexible modeling of event sequences in diverse fields.

In this work, inspired by Hugging Face (Wolf et al., 2020) for computer vision and natural language processing, we take the initiative to build a central library, namely EasyTPP, of popular research assets (e.g., data, models, evaluation methods, documentations) with the following distinct merits:

1. **Standardization**. We establish a standardized benchmark to enable transparent comparison of models. Our benchmark currently hosts 5 popularly-used real-world datasets that cover diverse real-world domains (e.g., commercial, social), and will include datasets in other domains (e.g., earthquake and volcano eruptions). One of our contributions is to develop a unified format for these datasets and provide source code (with thorough documentation) for data processing. This effort will free future researchers from large amounts of data-processing work, and facilitate exploration in new research topics such as transfer learning and adaptation (see Section 6).

2. **Comprehensiveness**. Our second contribution is to provide a wide range of easy-to-use evaluation programs, covering popular evaluation metrics (e.g., log-likelihood, kinds of next-event prediction accuracies and sequence similarities) and significance tests (e.g., permutation tests). By using this shared set of evaluation programs, researchers in this area will not only achieve a higher pace of development, but also ensure a better reproducibility of their results.

3. **Convenience**. Another contribution of EasyTPP is a rich suite of modules (functions and classes) which will significantly facilitate future method development. We reproduced previous most-cited and competitive models by composing these modules like building LEGOs; other researchers can reuse the modules to build their new models, significantly accelerating their implementation and improving their development experience. Examples of modules are presented in section 3.

4. **Flexibility**. Our library is compatible with both PyTorch (Paszke et al., 2019) and TensorFlow (Abadi et al., 2016), the top-2 popular deep learning frameworks, and thus offers a great flexibility for future research in method development.

5. **Extensibility**. Following our documentation and protocols, one could easily extend the EasyTPP library by adding new datasets, new modules, new models, and new evaluation programs. This high extensibility will contribute to building a healthy open-source community, eventually benefiting the research area of event sequence modeling.

## 2 BACKGROUND

**Definition.** Suppose we are given a fixed time interval $[0, T]$ over which an event sequence is observed. Suppose there are $I$ events in the sequence at times $0 < t_1 < \ldots < t_I \leq T$. We denote the sequence as $x_{[0,T]} = (t_1, k_1), \ldots, (t_I, k_I)$ where each $k_i \in \{1, \ldots, K\}$ is a discrete event type. Note that representations in terms of time $t_i$ and the corresponding inter-event time $\tau_i = t_i - t_{i-1}$ are isomorphic, we use them interchangeably. TPPs are probabilistic models for such event sequences. If we use $p_k(t \mid x_{[0,t)})$ to denote the probability that an event of type $k$ occurs over the infinitesimal interval $[t, t+dt)$, then the probability that nothing occurs will be $1 - \sum_{k=1}^{K} p_k(t \mid x_{[0,t)})$. Formally, the distribution of a TPP can be characterized by the **intensity** $\lambda_k(t \mid x_{[0,t)}) \geq 0$ for each event type $k$ at each time $t > 0$ such that $p_k(t \mid x_{[0,t)}) = \lambda_k(t \mid x_{[0,t)})dt$.

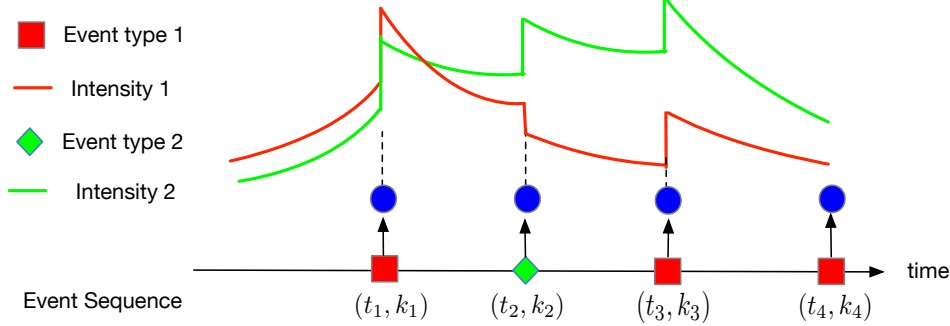

Figure 2: Drawing an event stream from a neural TPP. The model reads the sequence of past events (polygons) to arrive at a hidden state (blue). That state determines the future "intensities" of the two types of events–that is, their time-varying instantaneous probabilities. The intensity functions are continuous parametric curves (solid lines) determined by the most recent model state. Events will update the future intensity curves as they occur.

**Neural TPPs.** A neural TPP model autoregressively generates events one after another via neural networks. A schematic example is shown in Figure 2 and a detailed description on data samples can be found at the documentation of the repository. For the $i$-th event $(t_i, k_i)$, it computes the embedding of the event $\boldsymbol{e}_i \in \mathbb{R}^D$ via an embedding layer and the hidden state $\boldsymbol{h}_i$ gets updated conditioned on $\boldsymbol{e}_i$ and the previous state $\boldsymbol{h}_{i-1}$. Then one can draw the next event conditioned on the hidden state $\boldsymbol{h}_i$:

$$t_{i+1}, k_{i+1} \sim \mathbb{P}_\theta(t_{i+1}, k_{i+1} | \boldsymbol{h}_i), \quad \boldsymbol{h}_i = f_{update}(\boldsymbol{h}_{i-1}, \boldsymbol{e}_i), \tag{1}$$

where $f_{update}$ denotes a recurrent encoder, which could be either RNN (Du et al., 2016; Mei & Eisner, 2017) or more expressive attention-based recursion layer (Zhang et al., 2020; Zuo et al., 2020; Yang et al., 2022). A new line of research models the evolution of the states completely in continuous time:

$$\boldsymbol{h}_{i-} = f_{evo}(\boldsymbol{h}_{i-1}, t_{i-1}, t_i) \quad \text{between event times} \tag{2}$$
$$\boldsymbol{h}_i = f_{update}(\boldsymbol{h}_{i-}, \boldsymbol{e}_i) \quad \text{at event time } t_i \tag{3}$$

The state evolution in Equation (2) is generally governed by an ordinary differential equation (ODE) (Rubanova et al., 2019). For a broad and fair comparison, in EasyTPP, we implement not only recurrent TPPs but also an ODE-based continuous-time state model.

**Learning TPPs.** Negative log-likelihood (NLL) is the default training objective for both classical and neural TPPs. The NLL of a TPP given the entire event sequence $x_{[0,T]}$ is

$$\sum_{i=1}^{I} \log \lambda_{k_i}(t_i \mid x_{[0,t_i)}) - \int_{t=0}^{T} \sum_{k=1}^{K} \lambda_k(t \mid x_{[0,t)}) dt \tag{4}$$

Derivations of this formula can be found in previous work (Hawkes, 1971; Mei & Eisner, 2017).

## 3 THE BENCHMARKING PIPELINE

Figure 3 presents the open benchmarking pipeline implemented in EasyTPP. This pipeline will facilitate future research in this area and help promote reproducible work. In this section, we introduce and discuss each of the key components.

**Data Preprocessing.** Following common practices, we split the set of sequences into disjoint train, validation, and test set. Normally, the input is fed batch-wise into the model; there may exist sequences of events that have unequal-length in the same batch. To feed the sequences of varying lengths into the model, we pad all sequences to the same length, then use the "sequence_mask" tensor to identify which event tokens are padding. As we implemented several variants of attention-based TPPs, we also generated the "attention_mask" to mask all the future positions at each event

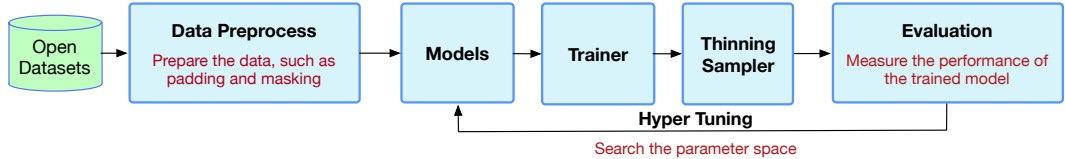

Figure 3: An open benchmarking pipeline using EasyTPP.

to avoid "peeking into the future". The padding and masking mechanism is the same as that used in NLP filed. See Appendix C.1 for a detailed explanation on sequence padding and masking in EasyTPP.

**Model Implementation.** Our EasyTPP library provides a suite of modules, and one could easily build complex models by composing these modules. Specifically, we implemented the models (see section 5.1) evaluated in this paper with our suite of modules (e.g., continuous-time LSTM, continuous-time attention). Moreover, some modules are model-agnostic methods for training and inference, which will further speed up the development speed of future methodology research. Below are two signature examples:

- compute_loglikelihood (function), which calculates log-likelihood of a model given data. It is non-trivial to correctly implement it due to the integral term of log-likelihood in Equation (4), and we have found errors in popular implementations.
- EventSampler (class), which draws events from a given point process via the thinning algorithm. The thinning algorithm is commonly used in inference but it is non-trivial to implement (and rare to see) an efficient and batched version. Our efficient and batched version (which we took great efforts to implement) will be useful for nearly all intensity-based event sequence models.

**Training.** We can estimate the model parameters by locally maximizing the NLL in Equation (4) with any stochastic gradient method. Note that computing the NLL can be challenging due to the presence of the integral in the second term in Equation (4). In EasyTPP, by default, we approximate the integral by Monte-Carlo estimation to compute the overall NLL (see Appendix B.1). Nonetheless, EasyTPP also incorporates intensity-free models (Shchur et al., 2020), whose objectives are easier to compute.

**Sampling.** Given the learned parameters, we apply the minimum Bayes risk (MBR) principle to predict the time and type with the lowest expected loss. A recipe can be found in Appendix B.2. Note that other methods exist for predicting with a TPP, such as adding an MLP layer to directly output the time and type prediction (Zuo et al., 2020; Zhang et al., 2020). However, the primary focus of this benchmark is generative models of event sequences, for which the principal approach is the thinning algorithm (Ogata, 1988). In EasyTPP, we implemented a batched version of thinning algorithm, which is then used to evaluate the TPPs in our experiments.

**Hyperparameter Tuning.** Our EasyTPP benchmark provides programs for automatic hyperparameter tuning. In addition to classical grid search, we also integrate *Optuna* (Akiba et al., 2019) in our framework to adaptively prune the search grid.

## 4    SOFTWARE INTERFACE

The purpose of building EasyTPP is to provide a simple and standardized framework to allow users to apply different state-of-the-art (SOTA) TPPs to any datasets as they would like to. For researchers, EasyTPP provides an implementation interface to integrate new recourse methods in an easy-to-use way, which allows them to compare their method to already existing methods. See the pseudo implementation in listing 1. For industrial practitioners, the availability of benchmarking code helps them easily assess the applicability of TPP models for their own problems. See an example of running an existed model in EasyTPP in listing 2. The full documentation of software interfaces

can be found at our repo. For more details of software architecture, please see introduction part of online documentation and Appendix A as well.

# 5 EXPERIMENTAL EVALUATION

## 5.1 EXPERIMENTAL SETUP

We comprehensively evaluate 9 models in our benchmark, which include the classical **Multivariate Hawkes Process (MHP)** with an exponential kernel, (seeAppendix B for more details), and 8 widely-cited state-of-the-art neural models:

- Two RNN-based models: **Recurrent marked temporal point process (RMTPP)** (Du et al., 2016) and **neural Hawkes Process (NHP)** (Mei & Eisner, 2017).
- Three attention-based models: **self-attentive Hawkes pocess (SAHP)** (Zhang et al., 2020), **transformer Hawkes process (THP)** (Zuo et al., 2020), **attentive neural Hawkes process (AttNHP)** (Yang et al., 2022).
- One TPP with the fully neural network based intensity: **FullyNN** (Omi et al., 2019).
- One intensity-free model **IFTPP** (Shchur et al., 2020).
- One TPP with the hidden state evolution governed by a neural ODE: **ODETPP**. It is a simplified version of the TPP proposed by Chen et al. (2021) by removing the spatial component. .

```python
from easy_tpp.model.torch_model.
torch_basemodel import TorchBaseModel

# Custom TPP implementations need to
inherit from the BaseModel interface
class NewModel(TorchBaseModel):
    def __init__(self, model_config):
        super(NewModel, self).__init__(
model_config)

    # Forward along the sequence,
output the states / intensities at
event times
    def forward(self, batch):
        .
        .
        .
        return states

    # Compute the loglikelihood loss
    def loglike_loss(self, batch):
        .
        .
        .
        return loglike

    # Compute the intensities at given
sampling times, used by Thinning
sampler
    def
compute_intensities_at_sample_times(self
, batch, sample_times, **kwargs):
        .
        .
        .
        return intensities
```

Listing 1: Pseudo implementation of customizing a TPP model in PyTorch using EasyTPP.

```python
import argparse
from easy_tpp.config_factory import
Config
from easy_tpp.runner import Runner

def main():
    parser = argparse.ArgumentParser()
    parser.add_argument('--config_dir',
                        type=str,
                        required=True,
                        help='Dir of
config to train and evaluate the model
.')
    parser.add_argument('--experiment_id
',
                        type=str,
                        required=True,
                        help='
Experiment id in the config file.')

    args = parser.parse_args()

    # Build up the configuation
    config = Config.build_from_yaml_file
(args.config_dir, args.experiment_id)

    # Intialize the runner of pipeline
    model_runner = Runner.
build_from_config(config)

    # Start running
    model_runner.run()

if __name__ == '__main__':
    main()
```

Listing 2: Example implementation of running a TPP model using EasyTPP.

We conduct experiments on 1 synthetic and 5 real-world datasets from popular works that contain diverse characteristics in terms of their application domains and temporal statistics (see Table 2):

- **Synthetic**. This dataset contains synthetic event sequences from a univariate Hawkes process sampled using Tick (Bacry et al., 2017) whose conditional intensity function is defined by $\lambda(t) = \mu + \sum_{t_i < t} \alpha\beta \cdot \exp\left(-\beta(t - t_i)\right)$ with $\mu = 0.2, \alpha = 0.8, \beta = 1.0$. We randomly sampled disjoint train, dev, and test sets with $1200, 200$ and $400$ sequences.

- **Amazon**(Ni, 2018). This dataset includes time-stamped user product reviews behavior from January, 2008 to October, 2018. Each user has a sequence of produce review events with each event containing the timestamp and category of the reviewed product, with each category corresponding to an event type. We work on a subset of $5200$ most active users with an average sequence length of $70$ and then end up with $K = 16$ event types.

- **Retweet** (Ke Zhou & Song, 2013). This dataset contains time-stamped user retweet event sequences. The events are categorized into $K = 3$ types: retweets by "small," "medium" and "large" users. Small users have fewer than 120 followers, medium users have fewer than 1363, and the rest are large users. We work on a subset of $5200$ active users with an average sequence length of $70$.

- **Taxi** (Whong, 2014). This dataset tracks the time-stamped taxi pick-up and drop-off events across the five boroughs of the New York City; each (borough, pick-up or drop-off) combination defines an event type, so there are $K = 10$ event types in total. We work on a randomly sampled subset of $2000$ drivers with an average sequence length of $39$.

- **Taobao** (Xue et al., 2022). This dataset contains time-stamped user click behaviors on Taobao shopping pages from November 25 to December 03, 2017. Each user has a sequence of item click events with each event containing the timestamp and the category of the item. The categories of all items are first ranked by frequencies and the top 19 are kept while the rest are merged into one category, with each category corresponding to an event type. We work on a subset of $4800$ most active users with an average sequence length of $150$ and then end up with $K = 20$ event types.

- **StackOverflow** (Leskovec & Krevl, 2014). This dataset has two years of user awards on a question-answering website: each user received a sequence of badges and there are $K = 22$ different kinds of badges in total. We work on a subset of $2200$ active users with an average sequence length of $65$.

**Evaluation Protocol.** We keep the model architectures as the original implementations in their papers. For a fair comparison, we use the same training procedure for all the models: we used Adam (Kingma & Ba, 2015) with the default parameters, biases initialized with zeros, no learning rate decay, the same maximum number of training epochs, and early stopping criterion (based on log-likelihood on the held-out dev set) for all models.

We mainly examine the models in two standard scenarios.

- Goodness-of-fit: we fit the models on the train set and measure the log-probability they assign to the held-out data.

- Next-event prediction: we use the minimum Bayes risk (MBR) principle to predict the next event time given only the preceding events, as well as its type given both its true time and the preceding events. We evaluate the time and type prediction by RMSE and error rate, respectively.

In addition, we propose a new evaluation task: the long-horizon prediction. Given the prefix of each held-out sequence $x_{[0,T]}$, we autoregressively predict the next events in a future horizon $\hat{x}_{(T,T']}$. It is evaluated by measuring the optimal transport distance (OTD), a type of edit distance for event sequences (Mei et al., 2019), between the prediction $\hat{x}_{(T,T']}$ and ground truth $x_{(T,T']}$. As pointed out by Xue et al. (2022), long-horizon prediction of event sequences is essential in various real-world domains, and this task provides new insight into the predictive performance of the models.

It is worth noting that the original version of the FullyNN model does not support multi-type event sequences. Therefore it is excluded from the type prediction task.

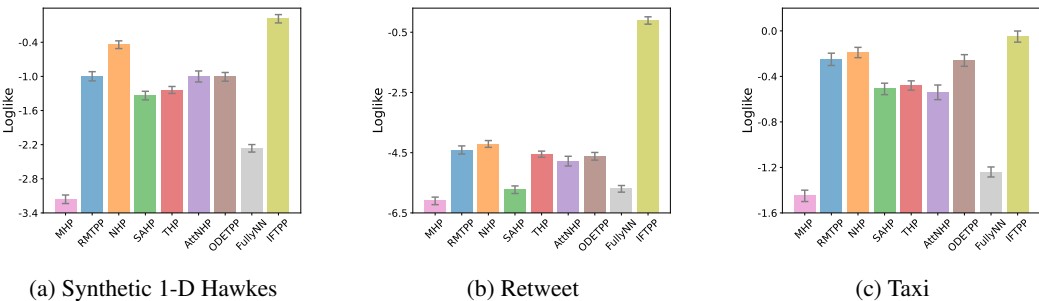

| (a) Synthetic 1-D Hawkes | (b) Retweet | (c) Taxi |

Figure 4: Performance of all the methods on the goodness-of-fit task on synthetic Hawkes, Retweet, and Taxi data. A higher score is better. All methods are implemented in PyTorch.

| MODEL | METRICS (TIME RMSE / TYPE ERROR RATE) | | | | |
|---|---|---|---|---|---|
| | AMAZON | RETWEET | TAXI | TAOBAO | STACKOVERFLOW |
| MHP | 0.635/75.9% | 22.92/55.7% | 0.382/9.53% | 0.539/68.1% | 1.388/65.0% |
| | 0.005/0.005 | 0.212/0.004 | 0.002/0.0004 | 0.004/0.004 | 0.011/0.005 |
| RMTPP | 0.620/68.1% | 22.31/44.1% | 0.371/9.51% | 0.531/55.8% | 1.376/57.3% |
| | 0.005/0.006 | 0.209/0.003 | 0.003/0.0003 | 0.005/0.004 | 0.018/0.005 |
| NHP | 0.621/67.1% | 21.90/40.0% | 0.369/8.50% | 0.531/54.2% | 1.372/55.0% |
| | 0.005/0.006 | 0.184/0.002 | 0.003/0.0005 | 0.005/0.006 | 0.011/0.006 |
| SAHP | 0.619/67.7% | 22.40/41.6% | 0.372/9.75% | 0.532/54.6% | 1.375/56.1% |
| | 0.005/0.006 | 0.301/0.002 | 0.003/0.0008 | 0.004/0.002 | 0.013/0.005 |
| THP | 0.621/66.1% | 22.01/41.5% | 0.370/8.68% | 0.531/53.6% | 1.374/55.0% |
| | 0.003/0.007 | 0.188/0.003 | 0.003/0.0006 | 0.003/0.004 | 0.021/0.006 |
| ATTNHP | 0.621/65.3% | 22.19/40.1% | 0.371/8.71% | 0.529/53.7% | 1.372/55.2% |
| | 0.005/0.006 | 0.180/0.003 | 0.003/0.0004 | 0.005/0.001 | 0.019/0.003 |
| ODETPP | 0.620/65.8% | 22.48/43.2% | 0.371/10.54% | 0.533/55.4% | 1.374/56.8% |
| | 0.006/0.008 | 0.175/0.004 | 0.003/0.0008 | 0.005/0.007 | 0.022/0.004 |
| FULLYNN | 0.615/N.A. | 21.92/N.A. | 0.373/N.A. | 0.529/N.A. | 1.375/N.A. |
| | 0.005/N.A. | 0.159/N.A. | 0.003/N.A. | 0.005N.A. | 0.015/N.A. |
| IFTPP | 0.618/67.5% | 22.18/39.7% | 0.377/8.56% | 0.531/55.4% | 1.373/55.1% |
| | 0.005/0.007 | 0.204/0.003 | 0.003/0.006 | 0.005/0.004 | 0.010/0.005 |

Table 1: Performance of all the methods on next-event's time prediction and next-event's type prediction on five real datasets (for each model, first row corresponds to the metrics value while second row corresponds to the standard deviation). Lower score is better. All methods are implemented in PyTorch. As clarified, FullyNN is not applicable for the type prediction tasks.

## 5.2 RESULTS AND ANALYSIS

Figure 4 (see Table 7 for exact numbers in the figure) reports the log-likelihood on three held-out datasets for all the methods. We find IFTPP outperforms all the competitors because it evaluates the log-likelihood in a close form while the others (RMTPP, NHP, THP, AttNHP, ODETPP) compute the intensity function via Monte Carlo integration, causing numerical approximation errors. FullyNN method, which also exactly computes the log-likelihood, has worse fitness than other neural competitors. As Shchur et al. (2020) points out, the PDF of FullyNN does not integrate to 1 due to a suboptimal choice of the network architecture, therefore causing a negative impact on the performance.

Table 1 reports the time and type prediction results on five real datasets. We find there is no single winner against all the other methods. Attention-based methods (SAHP, THP, AttNHP) generally perform better than or close to non-attention methods (RMTPP, NHP, ODETPP,FullyNN and IFTPP) on Amazon, Taobao, and Stackoverflow, while NHP is the winner on both Retweet and Taxi. We

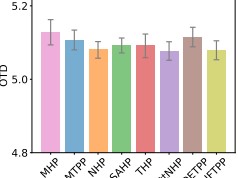 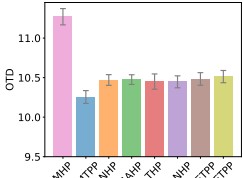 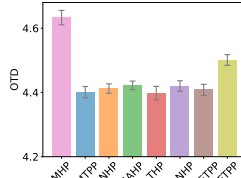 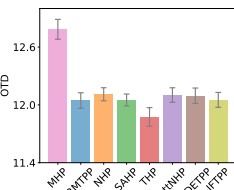

Figure 5: Long horizon prediction on Retweet data: left (avg prediction horizon 5 events) vs. right (avg prediction horizon 10 events).

Figure 6: Long horizon prediction on Taxi data: left (avg prediction horizon 5 events) vs. right (avg prediction horizon 10 events).

see that NHP is a comparably strong baseline with attention-based TPPs. This is not too surprising because similar results have been reported in previous studies (Yang et al., 2022).

Not surprisingly, the performance of the classical model MHP is worse than the neural models across most of the evaluation tasks, consistent with the previous findings that neural TPPs have demonstrated to be more effective than classical models at fitting data and making predictions.

Please see Appendix D.3 for more results (e.g., actual numbers of figures) on all the datasets. With a growing number of TPP methods proposed, we will continuously expand the catalog of models and datasets and actively update the benchmark in our repository.

**Analysis-I: Long Horizon Prediction.** We evaluate the long horizon prediction task on Retweet and Taxi datasets. On both datasets, we set the prediction horizon to be the one that approximately has 5 and 10 events, respectively. Shown in Figure 5 and Figure 6, we find that AttNHP and THP are two co-winners on Retweet and THP is a single winner on Taxi. Nonetheless, the margin of the winner over the competitors is small. The exact numbers shown in these two figures could be found in Table 5 in Appendix D.3. Because these models are autoregressive and locally normalized, they are all exposed to cascading errors. To fix this issue, one could resort to other kinds of models such as a hybridly normalized model (Xue et al., 2022), which is out of the scope of the paper.

**Analysis-II: Models with Different Frameworks—PyTorch vs. TensorFlow.** Researchers normally implement their experiments and models for specific ML frameworks. For example, recently proposed methods are mostly restricted to PyTorch and are not applicable to TensorFlow models. As explained in Section 4, to facilitate the use of TPPs, we implement two equivalent sets of methods in PyTorch and TensorFlow. Table 6 in Appendix D.3 shows the relative difference between the results of Torch and TensorFlow implementations are all within $[-1.5\%, 1.5\%]$. To conclude, the two sets of models produce similar performance in terms of predictive ability.

## 6 FUTURE RESEARCH OPPORTUNITIES

We summarize our thoughts on future research opportunities inspired by our benchmarking results.

Most importantly, the results seem to be signaling that we should think beyond architectural design. For the past decade, this area has been focusing on developing new architectures, but the performance of new models on the standard datasets seem to be saturating. Notably, all the best to-date models make poor predictions on time of future events. Moreover, on type prediction, attention-based model (Zuo et al., 2020; Zhang et al., 2020; Yang et al., 2022) only outperform other architectures by a small margin. Looking into the future, we advocate for a few new research directions that may bring significant contributions to the field.

The first is to build foundation models for event sequence modeling. The previous model-building work all learns data-specific weights, and does not test the transferring capabilities of the learned models. Inspired by the emergence of foundation models in other research areas, we think it will be beneficial to explore the possibility to build foundation models for event sequences. Conceptually, learning from a large corpus of diverse datasets—like how GPTs (Nakano et al., 2021) learn by reading open web text—has great potential to improve the model performance and generalization beyond what could be achieved in the current in-domain in-data learning paradigm. Our library can facilitate exploration in this direction since we unify the data formats and provide an easy-to-use

interface that users can seamlessly plug and play any set of datasets. Challenges in this direction arise as different datasets tend to have disjoint sets of event types and different scales of time units.

The second is to go beyond event data itself and utilize external sources to enhance event sequence modeling. Seeing the performance saturation of the models, we are inspired to think whether the performance has been bounded by the intrinsic signal-to-noise ratio of the event sequence data. Therefore, it seems natural and beneficial to explore the utilization of other information sources, which include but are not limited to: (i) sensor data such as satellite images and radiosondes signals; (ii) structured and unstructured knowledge bases (e.g., databases, Wikipedia); (iii) large pretrained models such as ChatGPT (Brown et al., 2020), may assist event sequence models in improving their prediction accuracies. Concurrent with this work, Shi et al. (2023) has made an early step in this direction.

The third is to go beyond observational data and embed event sequence models into real-world interventions (Qu et al., 2023). With interventional feedback from the real world, an event sequence model would have the potential to learn real causal dynamics of the world, which may significantly improve prediction accuracy.

All the aforementioned directions open up research opportunities for technical innovations.

## 7 RELATED WORK

**Temporal Point Processes.** Over recent years, a large variety of RNN-based TPPs have been proposed (Du et al., 2016; Mei & Eisner, 2017; Xiao et al., 2017; Omi et al., 2019; Shchur et al., 2020; Mei et al., 2020; Boyd et al., 2020). Models of this kind enjoy continuous state spaces and flexible transition functions, thus achieving superior performance on many real datasets, compared to the classical Hawkes process (Hawkes, 1971). To properly capture the long-range dependency in the sequence, the attention and transformer techniques (Vaswani et al., 2017) have been adapted to TPPs (Zuo et al., 2020; Zhang et al., 2020; Yang et al., 2022; Wen et al., 2023) and makes further improvements on predictive performance. There has also been research in creative ways of training temporal point processes, such as in a meta learning framework (Bae et al., 2023). Despite significant progress made in academia, the existing studies usually perform model evaluations and comparisons in an ad-hoc manner, e.g., by using different experimental settings or different ML frameworks. Such conventions not only increase the difficulty in reproducing these methods but also may lead to inconsistent experimental results among them.

**Open Benchmarking on TPPs.** The significant attention attracted by TPPs in recent years naturally leads to a high demand for an open benchmark to fairly compare against baseline models. While many efforts have been made in the domains of recommender systems (Zhu et al., 2021) and natural language processing (Wang et al., 2019), benchmarking TPPs is an under-explored topic. Tick (Bacry et al., 2017) and pyhawkes[1] are two well-known libraries that focus on statistical learning for classical TPPs, which are not suitable for the SOTA neural models. Poppy (Xu, 2018) is a PyTorch-based toolbox for neural TPPs, but it has not been actively maintained since 2021 and has not implemented any recent SOTA methods. To the best of our knowledge, EasyTPP is the first package that provides open benchmarking for popular neural TPPs.

## 8 CONCLUSION

In this work, we presented EasyTPP, an open and comprehensive benchmark for standardized and transparent comparison of TPP models. The benchmark hosts a diversity of datasets and models. In addition, it provides a user-friendly interface and a rich library, with which one could easily integrate new datasets and implement new models. With these features, EasyTPP has the potential to significantly facilitate future research in the area of event sequence modeling.

---

[1]https://github.com/slinderman/pyhawkes

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

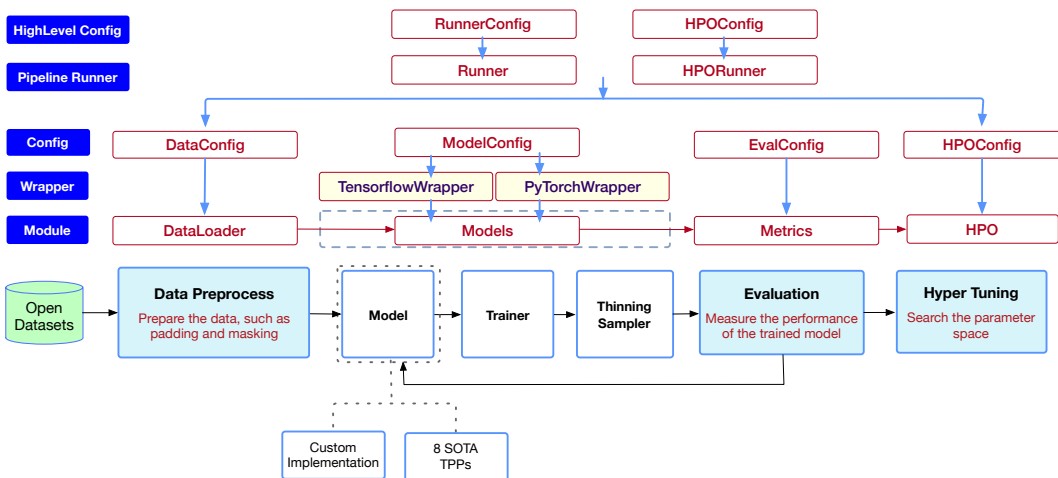

Figure 7: Architecture of the EasyTPP library. The dashed arrows show the different implementation possibilities, either to use pre-defined SOTA TPP models or provide a custom implementation. All dependencies between the configurations and modules are visualized by solid arrows with additional descriptions.

# Appendices

## A  EASYTPP'S SOFTWARE INTERFACE DETAILS

In this section, we describe the architecture of our open-source benchmarking software EasyTPP in more detail and provide examples of different use cases and their implementation.

### A.1  HIGH LEVEL SOFTWARE ARCHITECTURE

The purpose of building EasyTPP is to provide a simple and standardized framework to allow users to apply different state-of-the-art (SOTA) TPPs to arbitrary data sets. For researchers, EasyTPP provides an implementation interface to integrate new recourse methods in an easy-to-use way, which allows them to compare their method to already existing methods. For industrial practitioners, the availability of benchmarking code helps them easily assess the applicability of TPP models for their own problems.

A high level visualization of the EasyTPP's software architecture is depicted in Figure 7. *Data Preprocess* component provides a common way to access the event data across the software and maintains information about the features. For the *Model* component, the library provides the possibility to use existing methods or extend the users' custom methods and implementations. A *wrapper* encapsulates the black-box models along with the trainer and sampler. The primary purpose of the wrapper is to provide a common interface to easily fit in the training and evaluation pipeline, independently of their framework (e.g., PyTorch, TensorFlow). See Appendix A.2 and Appendix A.3 for details. The running of the pipeline is parameterized by the configuration class - *RunnerConfig* (without hyper-parameter tuning) and *HPOConfig* (with hyper-parameter tuning).

### A.2  WHY DOES EASYTPP SUPPORT BOTH TENSORFLOW AND PYTORCH

TensorFlow and PyTorch are the two most popular Deep Learning (DL) frameworks today. PyTorch has a reputation for being a research-focused framework, and indeed, most of the authors have implemented TPPs in PyTorch, which are used as references by EasyTPP. On the other hand, TensorFlow has been widely used in real world applications. For example, Microsoft recommender,[2] NVIDIA

---

[2]https://github.com/microsoft/recommenders.

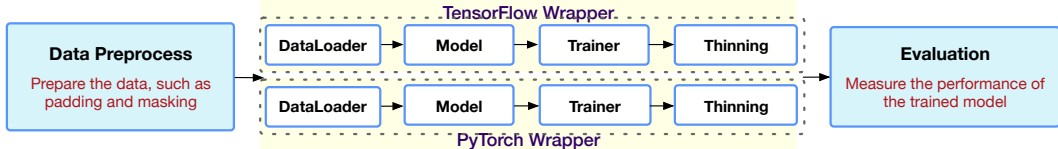

Figure 8: Illustration of TensorFlow and PyTorch Wrappers in the EasyTPP library.

Merlin[3] and Alibaba EasyRec[4] are well-known industrial user modeling systems with TensorFlow as the backend. In recent works, TPPs have been introduced to better capture the evolution of the user preference in continuous-time (Bao & Zhang, 2021; Fan et al., 2021; Bai et al., 2019). To support the use of TPPs by industrial practitioners, we implement an equivalent set of TPPs in TensorFlow. As a result, EasyTPP not only helps researchers analyze the strengths and bottlenecks of existing models, but also facilitates the deployment of TPPs in industrial applications.

## A.3 HOW DOES EASYTPP SUPPORT BOTH PYTORCH AND TENSORFLOW

We implement two equivalent sets of data loaders, models, trainers, thinning samplers in TensorFlow and PyTorch, respectively, then use wrappers to encapsulate them so that they have the same API exposed in the whole training and evaluation pipeline. See Figure 8.

## A.4 EASYTPP FOR RESEARCHERS

The research groups can inherit from the *BaseModel* to implement their own method in EasyTPP. This opens up a way of standardized and consistent comparisons between different TPPs when exploring new models.

Specifically, if we want to customize a TPP in PyTorch, we need to initialize the model by inheriting the class *TorchBaseModel*:

```python
from easy_tpp.model.torch_model.torch_basemodel import TorchBaseModel

# Custom Torch TPP implementations need to
# inherit from the TorchBaseModel interface
class NewModel(TorchBaseModel):
    def __init__(self, model_config):
        super(NewModel, self).__init__(model_config)

    # Forward along the sequence, output the states / intensities at the event
times
    def forward(self, batch):
        ...
        return states

    # Compute the loglikelihood loss
    def loglike_loss(self, batch):
        ....
        return loglike

    # Compute the intensities at given sampling times
    # Used in the Thinning sampler
    def compute_intensities_at_sample_times(self, batch, sample_times, **kwargs):
        ...
        return intensities
```

[3]https://developer.nvidia.com/nvidia-merlin.
[4]https://github.com/alibaba/EasyRec.

---

Listing 3: Pseudo implementation of customizing a TPP model in PyTorch using EasyTPP.

Equivalent, if we want to customize a TPP in TensorFlow, we need to initialize the model by inheriting the class *TfBaseModel*:

---

```python
from easy_tpp.model.torch_model.tf_basemodel import TfBaseModel

# Custom Torch TPP implementations need to
# inherit from the TorchBaseModel interface
class NewModel(TfBaseModel):
    def __init__(self, model_config):
        super(NewModel, self).__init__(model_config)

    # Forward along the sequence, output the states / intensities at the event
times
    def forward(self, batch):
        ...
        return states

    # Compute the loglikelihood loss
    def loglike_loss(self, batch):
        ....
        return loglike

    # Compute the intensities at given sampling times
    # Used in the Thinning sampler
    def compute_intensities_at_sample_times(self, batch, sample_times, **kwargs):
        ...
        return intensities
```

Listing 4: Pseudo implementation of customizing a TPP model in TensorFlow using EasyTPP.

## A.5 EasyTPP as a Modeling Library

A common usage of the package is to train and evaluate some standard TPPs. This can be done by loading black-box-models and data sets from our provided datasets, or by user-defined models and datasets via integration with the defined interfaces. Listing 5 shows an implementation example of a simple use-case, fitting a TPP model method to a preprocessed dataset from our library.

---

```python
import argparse

from easy_tpp.config_factory import Config
from easy_tpp.runner import Runner

def main():
    parser = argparse.ArgumentParser()

    parser.add_argument('--config_dir',
                        type=str,
                        required=False,
                        default='configs/experiment_config.yaml',
                        help='Dir of configuration yaml to train and evaluate the
 model.')

    parser.add_argument('--experiment_id',
                        type=str,
                        required=False,
                        default='IntensityFree_train',
                        help='Experiment id in the config file.')
```

```
    args = parser.parse_args()

    # Build up the configuation for the runner
    config = Config.build_from_yaml_file(args.config_dir, experiment_id=args.
experiment_id)

    # Intialize the runner for the pipeline
    model_runner = Runner.build_from_config(config)

    # Start running
    model_runner.run()

if __name__ == '__main__':
    main()
```

Listing 5: Example implementation of running a TPP model using EasyTPP.

## B  MODEL IMPLEMENTATION DETAILS

We have implemented the following TPPs

- **Multivariate Hawkes Process (MHP)**. We implemented it using Tick (Bacry et al., 2017) with an exponential kernel with fixed decays: the base intensity is set to 0.2 for all dimensions while the decay matrix is set to be a unit matrix plus a small noise. See online document for more details.

- **Recurrent marked temporal point process (RMTPP)** (Du et al., 2016). We implemented both the Tensorflow and PyTorch version of RMTPP by our own.

- **Neural Hawkes process (NHP)** (Mei & Eisner, 2017) and **Attentive neural Hawkes process (AttNHP)** (Yang et al., 2022). The Pytorch implementation mostly comes from the code from the public GitHub repository at `https://github.com/yangalan123/anhp-andtt` (Yang et al., 2022) with MIT License. We developed the Tensorflow version of NHP and ttNHP by our own.

- **Self-attentive Hawkes process (SAHP)** (Zhang et al., 2020) and **transformer Hawkes process (THP)** (Zuo et al., 2020). We rewrote the PyTorch versions of SAHP and THP based on the public Github repository at `https://github.com/yangalan123/anhp-andtt` (Yang et al., 2022) with MIT License. We developed the Tensorflow versions of the two models by our own.

- **Intensity-free TPP (IFTPP)** (Shchur et al., 2020). The Pytorch implementation mostly comes from the code from the public GitHub repository at `https://github.com/shchur/ifl-tpp` (Shchur et al., 2020) with MIT License. We implemented a Tensorflow version by our own.

- **Fully network based TPP (FullyNN)** (Omi et al., 2019). We rewrote both the Tensorflow and PyTorch versions of the model faithfully based on the author's code at `https://github.com/omitakahiro/NeuralNetworkPointProcess`. Please not that the model only considers the number of the types to be one, i.e., the sequence's $K = 1$.

- **ODE-based TPP (ODETPP)** (Chen et al., 2021). We implement a TPP model, in both Tensorflow and PyTorch, with a continuous-time state evolution governed by a neural ODE. It is basically the spatial-temporal point process (Chen et al., 2021) without the spatial component.

### B.1  LIKELIHOOD COMPUTATION DETAILS

In this section, we discuss the implementation details of NLL computation in Equation (4).

The integral term in Equation (4) is computed using the Monte Carlo approximation given by Mei & Eisner (2017, Algorithm 1), which samples times $t$. This yields an unbiased stochastic gradient. For the number of Monte Carlo samples, we follow the practice of Mei & Eisner (2017): namely, at training time, we match the number of samples to the number of observed events at training time, a reasonable and fast choice, but to estimate log-likelihood when tuning hyperparameters or reporting final results, we take 10 times as many samples.

At each sampled time $t$, the Monte Carlo method still requires a summation over all events to obtain $\lambda(t)$. This summation can be expensive when there are many event types. This is not a serious problem for our EasyTPP implementation since it can leverage GPU parallelism.

### B.2 NEXT EVENT PREDICTION

It is possible to sample event sequences exactly from any intensity-based model in EasyTPP, using the **thinning algorithm** that is traditionally used for autoregressive point processes (Lewis & Shedler, 1979; Liniger, 2009). In general, to apply the thinning algorithm to sample the next event at time $\geq t_0$, it is necessary to have an upper bound on $\{\lambda_e(t) : t \in [t_0, \infty)\}$ for each event type $t$. An explicit construction for the NHP (or AttNHP) model was given by Mei & Eisner (2017, Appendix B.3).

Section 3 includes a task-based evaluation where we try to predict the *time* and *type* of just the next event. More precisely, for each event in each held-out sequence, we attempt to predict its time given only the preceding events, as well as its type given both its true time and the preceding events.

We evaluate the time prediction with average $L_2$ loss (yielding a root-mean-squared error, or **RMSE**) and evaluate the argument prediction with average 0-1 loss (yielding an **error rate**).

Following Mei & Eisner (2017), we use the minimum Bayes risk (MBR) principle to predict the time and type with the lowest expected loss. For completeness, we repeat the general recipe in this section.

For the $i$-th event, its time $t_i$ has density $p_i(t) = \lambda(t)\exp(-\int_{t_{i-1}}^{t} \lambda(t')dt')$. We choose $\int_{t_{i-1}}^{\infty} tp_i(t)dt$ as the time prediction because it has the lowest expected $L_2$ loss. The integral can be estimated using i.i.d. samples of $t_i$ drawn from $p_i(t)$ by the thinning algorithm.

Given the next event time $t_i$, we choose the most probable type $\arg\max_e \lambda_e(t_i)$ as the type prediction because it minimizes expected 0-1 loss.

### B.3 LONG HORIZON PREDICTION

The TPP models are typically autoregressive: predicting each future event is conditioned on all the previously predicted events. Following the approach in (Xue et al., 2022), we set up a prediction horizon and use OTD to measure the divergence between the ground truth sequence and the predicted sequence within the horizon. For more details about the setup and evaluation protocol, please see Section 5 in Xue et al. (2022).

## C DATASET DETAILS

To comprehensively evaluate the models, we preprocessed one synthetic and five real-world datasets from widely-cited works that contain diverse characteristics in terms of their application domains and temporal statistics. All preprocessed datasets are available at Google Drive.

- **Synthetic.** This dataset contains synthetic event sequences from a univariate Hawkes process sampled using Tick (Bacry et al., 2017) whose conditional intensity function is defined by

$$\lambda(t) = \mu + \sum_{t_i < t} \alpha\beta \cdot \exp(-\beta(t - t_i))$$

  with $\mu = 0.2, \alpha = 0.8, \beta = 1.0$. We randomly sampled disjoint train, dev, and test sets with 1200, 200 and 400 sequences.

- **Amazon** (Ni, 2018). This dataset includes time-stamped user product reviews behavior from January, 2008 to October, 2018. Each user has a sequence of produce review events with each event containing the timestamp and category of the reviewed product, with each category corresponding to an event type. We work on a subset of 5200 most active users with an average sequence length of 70 and then end up with $K = 16$ event types.

- **Retweet** (Ke Zhou & Song, 2013). This dataset contains time-stamped user retweet event sequences. The events are categorized into $K = 3$ types: retweets by "small," "medium" and

| DATASET | $K$ | # OF EVENT TOKENS | | | SEQUENCE LENGTH | | |
|---|---|---|---|---|---|---|---|
| | | TRAIN | DEV | TEST | MIN | MEAN | MAX |
| RETWEET | 3 | 369000 | 62000 | 61000 | 10 | 41 | 97 |
| TAOBAO | 17 | 350000 | 53000 | 101000 | 3 | 51 | 94 |
| AMAZON | 16 | 288000 | 12000 | 30000 | 14 | 44 | 94 |
| TAXI | 10 | 51000 | 7000 | 14000 | 36 | 37 | 38 |
| STACKOVERFLOW | 22 | 90000 | 25000 | 26000 | 41 | 65 | 101 |
| HAWKES-1D | 1 | 55000 | 7000 | 15000 | 62 | 79 | 95 |

Table 2: Statistics of each dataset.

"large" users. Small users have fewer than 120 followers, medium users have fewer than 1363, and the rest are large users. We work on a subset of 5200 most active users with an average sequence length of 70.

- **Taxi** (Whong, 2014). This dataset tracks the time-stamped taxi pick-up and drop-off events across the five boroughs of the New York City; each (borough, pick-up or drop-off) combination defines an event type, so there are $K = 10$ event types in total. We work on a randomly sampled subset of 2000 drivers and each driver has a sequence. We randomly sampled disjoint train, dev and test sets with 1400, 200 and 400 sequences.

- **Taobao** (Xue et al., 2022). This dataset contains time-stamped user click behaviors on Taobao shopping pages from November 25 to December 03, 2017. Each user has a sequence of item click events with each event containing the timestamp and the category of the item. The categories of all items are first ranked by frequencies and the top 19 are kept while the rest are merged into one category, with each category corresponding to an event type. We work on a subset of 4800 most active users with an average sequence length of 150 and then end up with $K = 20$ event types.

- **StackOverflow** (Leskovec & Krevl, 2014). This dataset has two years of user awards on a question-answering website: each user received a sequence of badges and there are $K = 22$ different kinds of badges in total. We randomly sampled disjoint train, dev and test sets with 1400, 400 and 400 sequences from the dataset.

Table 2 shows statistics about each dataset mentioned above.

### C.1 PADDING AND MASKING

Given an input sequence $x_{[0,T]} = (t_1, k_1), \ldots, (t_I, k_I)$, for each event, we firstly use an embedding layer to map the event type $k_i$ of each event to a dense vector in higher space; then pass it to the following modules to construct the state embedding $h_i$.

Normally, the input is fed batch-wise into the model; there may exist sequences of events that have unequal-length in the same batch. We pad them to the same length, where the padding mechanism is the same as that in NLP when we pad text sentences.

1. Users can choose to decide whether padding to the beginning or end, padding to max length of batch or a fixed length across the dataset. The function interface and usage are almost the same as those in huggingface/transformers package.

2. The sequence_mask is used to denote whether the event is a padded one or a real one, which will be used in computation of log-likelihood and type/rmse evaluation as well.

A more detailed explanation of dataset preprocessing operation can be found at our online document.

## D EXPERIMENT DETAILS

### D.1 SETUP

**Training Details.** For TPPs, the main hyperparameters to tune are the hidden dimension $D$ of the neural network and the number of layers $L$ of the attention structure (if applicable). In practice,

| MODEL | DESCRIPTION | VALUE USED |
|---|---|---|
| | *hidden_size* | 32 |
| | *time_emb_size* | 16 |
| RMTPP | *num_layers* | 2 |
| | *hidden_size* | 64 |
| | *time_emb_size* | 16 |
| NHP | *num_layers* | 2 |
| | *hidden_size* | 32 |
| | *time_emb_size* | 16 |
| SAHP | *num_layers* | 2 |
| | *num_heads* | 2 |
| | *hidden_size* | 64 |
| | *time_emb_size* | 16 |
| THP | *num_layers* | 2 |
| | *num_heads* | 2 |
| | *hidden_size* | 32 |
| | *time_emb_size* | 16 |
| ATTNHP | *num_layers* | 1 |
| | *num_heads* | 2 |
| | *hidden_size* | 32 |
| ODETPP | *time_emb_size* | 16 |
| | *num_layers* | 2 |
| | *hidden_size* | 32 |
| FULLYNN | *time_emb_size* | 16 |
| | *num_layers* | 2 |
| | *hidden_size* | 32 |
| INTENSITYFREE | *time_emb_size* | 16 |
| | *num_layers* | 2 |

Table 3: Descriptions and values of hyperparameters used for models.

the optimal $D$ for a model was usually $16, 32, 64$; the optimal $L$ was usually $1, 2, 3, 4$. To train the parameters for a given generator, we performed early stopping based on log-likelihood on the held-out dev set. The chosen parameters for the main experiments are given in Table 3.

**Computation Cost.** All the experiments were conducted on a server with 256G RAM, a 64 logical cores CPU (Intel(R) Xeon(R) Platinum 8163 CPU @ 2.50GHz) and one NVIDIA Tesla P100 GPU for acceleration. For training, the batch size is 256 by default. On all the dataset, the training of AttNHP takes most of the time (i.e., around 4 hours) while other models take less than 2 hours.

## D.2   SANITY CHECK

For each model we reproduced in our library, we ran experiments to ensure that our implementation could match the results in the original paper. We used the same hyperparameters as in original papers; we reran each experiment 5 times and took the average.

In Table 4, we show the relative differences between the implementations on Retweet and Taxi datasets. As we can see, all the relative differences are within $(-5\%, 5\%)$, indicating that our implementation is close to the original.

## D.3   MORE RESULTS.

For better visual comparisons, we present the results in Figure 4, Figure 5 and Figure 6 also in the form of tables, see Table 7 and Table 5.

The relative difference between the results of Torch and TensorFlow implementations can be found in Table 6.

| MODEL | METRICS (TIME RMSE / TYPE ERROR RATE) | |
| --- | --- | --- |
| | RETWEET | TAXI |
| RMTPP | $-4.1\%/-3.5\%$ | $-2.9\%/-3.7\%$ |
| NHP | $+3.4\%/+3.1\%$ | $+2.6\%/+3.5\%$ |
| SAHP | $+1.3\%/+1.7\%$ | $+1.1\%/+1.2\%$ |
| THP | $+1.3\%/+1.8\%$ | $-1.6\%/+1.5\%$ |
| ATTNHP | $+1.2\%/-1.0\%$ | $-1.2\%/-1.2\%$ |
| ODETPP | $-4.0\%/-3.9\%$ | $-4.3\%/-4.5\%$ |
| FULLYNN | $-5.0\%$/N.A. | $-4.1\%$/N.A. |
| IFTPP | $+3.4\%/+3.1\%$ | $+3.9\%/+3.0\%$ |

Table 4: The relative difference between the results of EasyTPP and original implementations.

| MODEL | OTD | | | |
| --- | --- | --- | --- | --- |
| | RETWEET AVG 5 EVENTS | RETWEET AVG 10 EVENTS | TAXI AVG 5 EVENTS | TAXI AVG 10 EVENTS |
| MHP | 5.128 (0.040) | 11.270 (0.091) | 4.633 (0.037) | 12.784 (0.111) |
| RMTPP | 5.107 (0.041) | 10.255 (0.099) | 4.401 (0.030) | 12.045 (0.114) |
| NHP | 5.080 (0.042) | 10.470 (0.085) | 4.412 (0.032) | 12.110 (0.125) |
| SAHP | 5.092 (0.039) | 10.475 (0.079) | 4.422 (0.039) | 12.051 (0.139) |
| THP | 5.091 (0.052) | 10.450 (0.090) | 4.398 (0.041) | 11.875 (0.108) |
| ATTNHP | 5.077 (0.039) | 10.447 (0.090) | 4.420 (0.044) | 12.102 (0.109) |
| ODETPP | 5.115 (0.041) | 10.483 (0.088) | 4.408 (0.039) | 12.095 (0.100) |
| FULLYNN | N.A. N.A. | N.A. N.A. | N.A. N.A. | N.A. N.A. |
| IFTPP | 5.079 (0.061) | 10.513 (0.115) | 4.501 (0.032) | 12.052 (0.121) |

Table 5: Long horizon prediction on Retweet and Taxi data (the first row is the prediction value while the second row is the standard deviation).

# E  ADDITIONAL NOTE

## E.1  CITATION COUNT IN ARXIV

We search the TPP-related articles in ArXiv `https://arxiv.org/` using their own search engine in three folds:

- Temporal point process: we search through the abstract of articles which contains the term 'temporal point process'.
- Hawkes process: we search through the abstract of articles with the term 'hawkes process' but without the term 'temporal point process'.
- Temporal event sequence: we search through the abstract of articles which include the term 'temporal event sequence' but exclude the term 'hawkes process' and 'temporal point process'.

We group the articles found out by the search engine by years and report it in Figure 1.

| MODEL | REL DIFF ON TIME RMSE (1ST ROW) AND TYPE ERROR RATE (2ND ROW) | | | | |
|---|---|---|---|---|---|
| | AMAZON | RETWEET | TAXI | TAOBAO | STACKOVERFLOW |
| RMTPP | −0.2% | +1.0% | +0.1% | +0.1% | +0.4% |
| | +0.5% | +1.3% | +0.6% | +0.2% | −0.7% |
| NHP | +0.7% | +0.5% | −0.2% | +0.1% | −0.1% |
| | +0.6% | +1.4% | +0.4% | −0.3% | −0.1% |
| SAHP | −0.8% | +0.7% | −0.8% | +0.4% | 0.3% |
| | +0.6% | +0.6% | −0.6% | +0.4% | 0.3% |
| THP | +0.6% | +0.6% | −0.2% | −0.5% | 0.6% |
| | +1.2% | +0.9% | −0.6% | +0.7% | 0.4% |
| ATTNHP | +0.4% | +0.4% | +0.3% | −0.1% | −0.2% |
| | +0.2% | −0.7% | −0.6% | +0.4% | +0.2% |
| ODETPP | −0.5% | +1.1% | +0.9% | +0.6% | 0.4% |
| | +0.8% | +1.3% | +1.1% | −0.5% | −0.5% |
| FULLYNN | +0.5% | −0.7% | −0.3% | −0.3% | +0.2% |
| | NA | NA | NA | NA | NA |
| IFTPP | −0.9% | +1.0% | +0.4% | +0.6% | +0.3% |
| | +0.4% | −0.7% | −0.3% | +0.2% | +0.2% |

Table 6: Relative difference between Torch and TensorFlow implementations of methods in Table 1.

| MODEL | METRICS (LOGLIKE) | | |
|---|---|---|---|
| | SYNTHETIC | RETWEET | TAXI |
| MHP | −3.150 | −5.949 | −1.466 |
| | (0.028) | (0.033) | (0.011) |
| RMTPP | −0.998 | −4.237 | −0.227 |
| | (0.009) | (0.033) | (0.001) |
| NHP | −0.443 | −4.137 | −0.208 |
| | (0.004) | (0.050) | (0.001) |
| SAHP | −1.337 | −5.009 | −0.478 |
| | (0.013) | (0.044) | (0.003) |
| THP | −1.238 | −4.560 | −0.442 |
| | (0.011) | (0.041) | (0.004) |
| ATTNHP | −1.001 | −4.756 | −0.491 |
| | (0.008) | (0.052) | (0.004) |
| ODETPP | −1.007 | −4.527 | −0.217 |
| | (0.007) | (0.044) | (0.002) |
| FULLYNN | −2.318 | −5.889 | −1.317 |
| | (0.014) | (0.032) | (0.009) |
| IFTPP | 0.186 | −0.212 | 0.019 |
| | (0.002) | (0.002) | (0.001) |

Table 7: Performance in numbers of all methods in
Figure 4 (the first row is the prediction value while the second row is the standard deviation).

## E.2 FUTURE DEVELOPMENT WORK

As the emergence of LLM-based temporal models (Zhou et al., 2023; Chu et al., 2023), we are making EasyTPP compatible with LLM models: such as hosting dataset / models in HuggingFace, building LLM-compatible training and evaluation pipelines.

We are committed to actively maintaining this benchmark and welcome contributions from other researchers and practitioners.

