# OpenReview forum: "EasyTPP: Towards Open Benchmarking Temporal Point Processes"
_ICLR.cc/2024/Conference — ICLR 2024 poster_

### Official Review · Reviewer_2AV3 · 2023-10-26

**Soundness:** 4 excellent
**Presentation:** 3 good
**Contribution:** 4 excellent
**Rating:** 8
**Confidence:** 4

**Summary:**

The authors propose an end-to-end benchmarking package for neural temporal point process models. The package contains 6 diverse real-world datasets, 1 synthetic dataset, 8 recent models, and support for both Tensorflow and PyTorch. The paper itself serves as a useful exposition of the field, tasks, types of models, proposed package, and detailed empirical results.

**Strengths:**

This paper is a welcome contribution to TPP modeling at a timely moment when significant neural point process approaches have been published yet many open questions still remain. The proposed package has all the hallmarks of usefulness:

1) Reproducible workflows via modular API.

2) Diverse models and datasets.

3) Supports top-2 ML languages equally (this is rare among benchmarking frameworks).

4) Open-source repository and datasets.

The paper is overall well-written.

**Weaknesses:**

There are some presentation/exposition gaps/weaknesses:

1) The accuracy plots would be more useful as tables.

2) The "temporal event sequence" topic is not well-defined.

3) The processing of sequence data is not adequately described.

I elaborate on these weaknesses in questions to the authors.

**Questions:**

My questions align with the three specific shortcomings I identified:

1) For reproducibility purposes, it would be useful to the community to know the exact accuracy values (and confidence bounds) from the various methods. Therefore in my opinion all plots showing empirical results should be replaced with tables. Is this possible in the camera-ready version?

2) I may have missed this in my reading, but in Fig 1, I don't quite know what is meant by "Temporal event sequence" as opposed to "temporal point process". Furthermore, isn't a Hawkes process a version of a temporal point process? It seems that these publication sets may overlap somewhat. I'm wondering if this is clarified in the text, or if the authors can clarify.

3) In Section 4's "Data Preprocess" section, the authors write "to feed the sequences of varying length into the model, we pad all sequences to the same length, then use the "sequence_mask" tensor to identify which even tokens are padding." These masks are not defined, and are referenced no where else in the manuscript, so in my opinion this passage carries almost no information. To my understanding (which may be incorrect), some TPP datasets consist of a single sequence, and some models may process the sequence recurrently, i.e. each state embedding h_t depends only on the previous embedding h_{t-1}. For these models, why are variable-length sequences needed? On the other hand, datasets with events of multiple types (and models that can handle multi-type events) will need variable-length sequences. It would be very useful to the reader if the authors could describe these multiple scenarios, and specifically how EasyTPP handles each one at the implementation level.

If given satisfactory answers to the above, I am willing to raise my score.

---

> ### Author Response · Authors · 2023-11-20
>
> > For reproducibility purposes, it would be useful to the community to know the exact accuracy values (and confidence bounds) from the various methods. Therefore in my opinion all plots showing empirical results should be replaced with tables. Is this possible in the camera-ready version?
>
> We agree that empirical results should be better illustrated with numbers instead of figures. We have replaced the figure of the main prediction result with a table in the main context and all the other tables are in Appendix D.3. We are committed to replace all the figures with tables in nice formats in the camera-ready version. See [Presentation Improvements].
>
> > I may have missed this in my reading, but in Fig 1, I don't quite know what is meant by "Temporal event sequence" as opposed to "temporal point process". Furthermore, isn't a Hawkes process a version of a temporal point process? It seems that these publication sets may overlap somewhat. I'm wondering if this is clarified in the text, or if the authors can clarify.
>
>
> In fact we have clarified this search process in Appendix E.1. In the revised version, we have added one line to refer to this section for details of how we count the articles.  See [Presentation Improvements].
>
> We clarify again here. To find out the number of point process research papers, we search in arxiv with the three terms (Temporal point process, Hawkes process,  Temporal event sequence) that are widely used in related publications, respectively. To avoid the overlap of searched results, we search with the following advanced rule:
> - Temporal point process: it corresponds to the articles whose abstracts contain the term ‘temporal point process’.
> - Hawkes process: it corresponds to the articles whose abstracts contain ‘hawkes process’ but not the term ‘temporal point process’.
> - Temporal event sequence: it corresponds to the articles whose abstracts contain the term ‘temporal event sequence’ without neither the term ‘hawkes process’ nor ‘temporal point process’.
>
>
>
> > In Section 4's "Data Preprocess" section... To my understanding (which may be incorrect), some TPP datasets consist of a single sequence, and some models may process the sequence recurrently, i.e. each state embedding h_t depends only on the previous embedding h_{t-1}. For these models, why are variable-length sequences needed? On the other hand, datasets with events of multiple types (and models that can handle multi-type events) will need variable-length sequences. It would be very useful to the reader if the authors could describe these multiple scenarios, and specifically how EasyTPP handles each one at the implementation level.
>
> Firstly, the dataset each contains thousands or tens of thousands of sequences, where each sequence usually corresponds to the sequence of one user’s behavior (e.g., one user’s reviewing behavior on Amazon). As a result, the variable-length sequence exists (not ‘needed’) in every dataset. See Appendix C for dataset descriptions and Table 2 for statistics of the dataset.
>
> Secondly, because the input is fed batch-wise into the model and there may exist sequences of events that have unequal-length in the same batch, we pad them to the same length batch-wise. This padding mechanism is the same as that in NLP when we pad text sentences.
> - Users can choose to decide whether padding to the beginning or end, padding to max length of batch or a fixed length across the dataset. The function interface and usage are almost the same as those in huggingface/transformers package.
> - The sequence_mask is used to denote whether the event is a padded one or a real one, which will be used in computation of log-likelihood and type/rmse evaluation as well.
> An detailed explanation of dataset preprocessing operation can be found at our documentation https://github.com/Anonymous0006/EasyTPP/blob/main/docs/source/user_guide/dataset.rst#expected-dataset-format-and-data-processing
>
> Lastly, to clarify, the input sequence format is the same no matter how many event types the data have, e.g., a sequence is $[(t_1, k_1), (t_2, k_2), (t_3, k_3),...]$, where $k_i$ is a non-negative integer denoting the event type. For all the models, we use an embedding layer (nn.Embedding in torch) to map the event type $k_i$ of each event to a dense vector in higher space; then pass it to the following modules to construct the state embedding $h_i$ recurrently (with $h_{t-1}$). We do not differentiate whether it is a single event type sequence or multi-type events because the codebase can handle both.
>
> The corresponding code of event type embedding can be found at https://github.com/Anonymous0006/EasyTPP/blob/main/easy_tpp/model/torch_model/torch_basemodel.py#L25.
>
> In sum, we have rephrased the paragraph and added Appendix C.1 for a more detailed explanation in the revised paper. See [Presentation Improvements].

---

> ### Author Response · Authors · 2023-11-22
> **A Reminder to Reviewer 2AV3**
>
> Dear Reviewer 2AV3,
>
> As a friendly reminder, we are waiting for your valuable feedback to our responses. In your summary of review, we appreciate recognizing the contribution of our work, and understand the concerns about clarity and presentation. We believe they have been addressed in the rebuttal and revised paper, so please check them out and let us know if you have any remaining questions or concerns. We are more than happy to discuss any further issues. We look forward to hearing from you. Thank you.

---

> > ### Comment · Reviewer_2AV3 · 2023-11-23
> > **Reply**
> >
> > Thanks to the authors for their replies and improvements. I have raised the presentation score by 1 and my overall score by 1.

---

> > > ### Author Response · Authors · 2023-11-23
> > >
> > > Thank you very much! We are excited to hear that our discussion has led to your continued positive evaluation of the work.

---

### Official Review · Reviewer_btMZ · 2023-10-30

**Soundness:** 3 good
**Presentation:** 2 fair
**Contribution:** 2 fair
**Rating:** 3
**Confidence:** 3

**Summary:**

To promote reproducible research in this field, this article proposes an open and central benchmark for point process research. The benchmark provides a complete pipeline and software interface. The effectiveness of this benchmark is demonstrated by the unified comparison of various model results.

**Strengths:**

1. The most important contribution lies in providing a simple and standardized framework to allow users
to apply different TPP models to any datasets, which promotes reproducible research in this field.

**Weaknesses:**

The main concern is contribution. Considering the quality of the ICLR community, although it is particularly urgent to propose a unified comparison and processing framework for point processes, the contribution is still limited.

**Questions:**

Would you mind elaborating further on the broader impact this benchmark will have on the community?

---

> ### Author Response · Authors · 2023-11-20
>
> > Would you mind elaborating further on the broader impact this benchmark will have on the community?
>
>
> Please see [Technical Contribution and Broader Impact]

---

> > ### Author Response · Authors · 2023-11-21
> > **Thank you for the review! Have we clearly addressed the concerns?**
> >
> > Dear Reviewer btMZ,
> >
> > We greatly appreciate the time you took to review our paper. In the rebuttal, we have clarified the contribution of our work and provided more detailed explanations for the proposed framework.
> >
> > Due to the short duration of the author-reviewer discussion phase, we would appreciate your feedback on whether your main concerns have been adequately addressed. We are ready and willing to provide further explanations and clarifications if necessary. Thank you very much!

---

> ### Author Response · Authors · 2023-11-23
> **A Reminder to Reviewer btMZ**
>
> As a reminder, the discussion session will end in few hours. We believe we have addressed your concerns on technical contribution.  If you have any further doubts or suggestions, we are willing to answer them until the last minutes. So, please let us know if our responses are satisfying or need more revision. We look forward to hearing from you. Thank you.

---

### Official Review · Reviewer_MAje · 2023-10-31

**Soundness:** 1 poor
**Presentation:** 1 poor
**Contribution:** 1 poor
**Rating:** 3
**Confidence:** 5

**Summary:**

This paper proposes a software package that abstracts the implementation of several temporal point process models.

**Strengths:**

1. Good motivation

**Weaknesses:**

1. Lack of technique contribution
2. Unclear presentation

**Questions:**

1. This paper lacks technical innovation. I don't think a software-level abstraction is a good fit for ICLR.
2. The description of the system design and interface lacks details.
3. It's unclear what points the authors intend to demonstrate in the experiment section.

---

> ### Author Response · Authors · 2023-11-20
>
> > This paper lacks technical innovation. I don't think a software-level abstraction is a good fit for ICLR.
>
> Please see [Technical Contribution and Broader Impact]
>
> > The description of the system design and interface lacks details.
>
>
> In section 3, we discuss the modules in detail and use figure-3 to illustrate the pipeline of the system.
>
> In section 4, we discuss the interface with two examples of usage (how to customize a model and how to run the pipeline)  in Listing 2 and Listing 3.
>
> In Appendix A, from page 13 - 16, we discuss the details of design and interface such as
> - High level software architecture (section A.1): figure-8 depicts the detailed architecture with each module’s functionality annotated / explained in text.
> - Why and how the system supports both Tensorflow and PyTorch optimization framework (section A.2 - A.3).
> - Pseudo implementation to customize a model using the interface of the code base (section A.4)
> - Pseudo implementation to run an existed model using the interface of the code base (section A.5)
>
> Besides, an even more detailed description of system and interface could be found at the document provided in our code base: https://github.com/Anonymous0006/EasyTPP/tree/main/docs/source/ref. Please see Readme.md to find out how to generate htmls for better visualization. The explanations of the interfaces at any level are well documented and easy to search and read.
>
> In all, around ¼ of the paper contents, plus pages of online documents,  relate to the system design and interface, could you please elaborate which part is still unclear to you？
>
> > It's unclear what points the authors intend to demonstrate in the experiment section.
>
> We want to demonstrate the following key insights from the experiment:
> - Insight 1: The performance of the classical model MHP is worse than the neural models across most of the evaluation tasks.
> - Insight 2: There is no single winner against all the other methods on time and type prediction. Notably, all the best to-date models make poor predictions on time of future events. Moreover, on type prediction, attention-based models only outperform other architectures by a small margin.
>
> These insights have strong implications on our benchmarking work and future research:
> - Based on insight 1, we decide that our code base focuses on neural TPPs rather than classical TPPs.
> - Insight 2 signals that we should think beyond architectural design for future directions. For the past decade, this area has been focusing on developing new architectures, but the performance of new models on the standard datasets seem to be saturating. Looking into the future, we advocate for a few new research directions that may bring significant contributions to the field, summarized in Section 6 [FUTURE RESEARCH OPPORTUNITIES], including building foundation models of event sequence, empowering the model with large language models etc.
>
> Please let me know if these points are clear for you.

---

> > ### Author Response · Authors · 2023-11-21
> > **Response to Reviewer MAje (before the end of discussion)**
> >
> > Dear Reviewer MAje,
> >
> > Since the End of author/reviewer discussions is just in one day, may we know if our response addresses your main concerns? If so, we kindly ask for your reconsideration of the score.
> >
> > Should you have any further advice on the paper and/or our rebuttal, please let us know and we will be more than happy to engage in more discussion and paper improvements. We would really appreciate it if our next round of communication could leave time for us to resolve any of your remaining or new questions.
> >
> > Thank you so much for devoting time to improving our benchmark!

---

> ### Author Response · Authors · 2023-11-23
> **A Reminder to Reviewer MAje**
>
> Dear Reviewer MAje,
>
> As the author-reviewer discussion is coming to its end in a few hours, we are looking forward to knowing whether our responses have resolved all your concerns so far, and whether you would like to adjust your evaluation. Thank you.

---

### Official Review · Reviewer_Xrns · 2023-11-03

**Soundness:** 4 excellent
**Presentation:** 3 good
**Contribution:** 4 excellent
**Rating:** 10
**Confidence:** 4

**Summary:**

The authors contribute the EasyTPP package for temporal point processes (TPPs). TPPs have been of increasing interest to the machine learning community in recent years, with many deep learning-based TPP models being able to outperform classic parametric TPP models in prediction tasks. There have not been many software packages implementing many different TPP models, which has made it difficult to compare and evaluate different models across standardized benchmark data sets and evaluation metrics. The EasyTPP package implements many recent TPP models and can be used with either PyTorch or TensorFlow. The authors utilize the package to benchmark these TPP models across a variety of data sets and prediction tasks.

*After author rebuttal:* I have read through the other reviews and author rebuttal and still strongly support the paper. I further expand on my reasoning in a reply to the author comment titled "Technical Contribution and Broader Impact."

**Strengths:**

- Addresses a major need in the machine learning community interested in TPPs--the lack of a standardized benchmarking tool. Most researchers are piecing together implementations from other papers in order to make comparisons. Existing software packages, such as PoPPy are out of date and not maintained. The proposed EasyTPP package could fill a major need for the community.
- Implements a comprehensive list of models and evaluation metrics, together with several representative data sets.
- The authors perform a thorough comparison of existing models using their EasyTPP package, which should also be useful to researchers in the area.

**Weaknesses:**

- The authors compare against the classical Multivariate Hawkes Process (MHP) but don't specify what type of kernel they use (I assume exponential) or structure of the excitation matrix.

Minor concerns:
- Some typos, e.g. Multivariate Hakwes Process (MHP)

**Questions:**

1. Is there an easy way to incorporate marks into your package? From the description in the paper, it seems like you can easily handle different event types through a multivariate TPP, but I don't see any discussion of marks. In many application settings, we can have a feature vector for each event, which can be modeled as a mark.

---

> ### Author Response · Authors · 2023-11-20
> **reply to Reviewer Xrns**
>
> > The authors compare against the classical Multivariate Hawkes Process (MHP) but don't specify what type of kernel they use (I assume exponential) or structure of the excitation matrix.
>
> We are indeed using the exponential kernel for MHP whose dimension equals the number of the event types of the dataset.
>
> We have added the specification for MHP in the revised version.  See [Presentation Improvements].
>
>
> > Some typos, e.g. Multivariate Hakwes Process (MHP)
>
> Thanks for pointing this out. We have fixed this typo in the revised version.  See [Presentation Improvements].
>
> > Is there an easy way to incorporate marks into your package? From the description in the paper, it seems like you can easily handle different event types through a multivariate TPP, but I don't see any discussion of marks. In many application settings, we can have a feature vector for each event, which can be modeled as a mark.
>
>
> Yes. It is easy to incorporate it in our package. We can modify the input data structure to include a feature vector for each event, and in the input module, we apply a feature encoder to produce a feature embedding and concat it with the event type embedding. This modification needs only a small amount of work and is compatible with the current framework.
>
> We are committed to adding this functionality in the future version. See [Commitment].

---

### Author Response · Authors · 2023-11-20
**Presentation Improvements**

We greatly appreciate the reviewers' insightful feedback on strengthening the presentation of our work. Based on the comments, we have made several key improvements to the paper:
- In Section 1, add a note to refer to appendix E.1 for more details of how the articles being counted in Figure 1.
- In Section 4, add url links of online documentation on introducing details of the software interface. We also emphasize that one can refer to Appendix A.1 for a much more detailed discussion on software architecture.
- In Section 3, we rephrase the data processing paragraph to address the padding mechanism used in EasyTPP and add Appendix C.1 for a more detailed explanation.
- In Section 5.1,  we add the description of the kernel for MHP (and a few lines in Appendix B as well) and fix typos pointed out by Reviewer Xrns.
- In Section 5.2,  in the main experiment, we replace figure 5 with table 1 that contains all the predictions of time and type with errs. Due to the page and time limit, we have put all the tables in Appendix D.3 (in fact the tables are already there in the original version) . We are committed to replace all the figures with well-formatted tables in the camera-ready version.

We have highlighted the change in green in the newly uploaded paper.

---

### Author Response · Authors · 2023-11-20
**Commitment**

We are fully committed to maintaining EasyTPP as a high-quality, community-driven resource that evolves with the field's progress. Specifically, we will:
- Modify the data processing and input modules per Reviewer Xrns's suggestion to incorporate additional vectors of features of events.
- Actively monitor new research advancements and regularly add datasets, models, evaluations, and other features to the library per user needs.

---

### Author Response · Authors · 2023-11-20
**Technical Contribution and Broader Impact**

Reviewer MAje is concerned with the technical innovation and doubts whether ``a software-level abstraction'' is fit for ICLR. Reviewer btMZ is also concerned about the contribution.

We submitted this paper to the 'datasets and benchmarks' track at ICLR because we believe it makes multiple technical innovations that constitute a novel contribution.  Isn't it a notable technical contribution to (be the first to) performing the open benchmarking of the time-stamped event sequences? Such data is ubiquitous in real-world applied domains (e.g., finance, medicine); no benchmarking work exists for such data. So why not have one? To summarize, we have taken the first step to address this need by:
- Carefully curating and preprocessing 5 popular real-world event sequence datasets from diverse applied domains like e-commerce and social networks. This enables standardized evaluation and frees future researchers from repetitive data wrangling.
- Implementing major neural TPP models like self-attention networks with a unified codebase and interface. This level of standardization did not exist before and will significantly improve reproducibility and ease future research.
- Providing a comprehensive suite of evaluation programs and metrics beyond log-likelihood, including next-event prediction and a new long-horizon assessment. Together with the datasets and models, these form an unprecedented open benchmark for fair comparisons.
- Thoroughly documenting the software usage and easy extensibility, allowing community contributions.


Critically, our work goes beyond software abstraction to also make broader impacts on TPP research:
- The unified data format opens new research directions like transfer learning and multi-task learning.
- The evaluation suite allows faster iteration and improves reproducibility. Researchers can easily build on it.
- Our analysis of the benchmark results motivates new research opportunities that are novel contributions, like building foundation models.
- We found some recent works in top conferences built upon our open-sourced codebase, showing EasyTPP's value to the community.

In summary, we believe the technical innovations above, especially creating the first open benchmark for neural TPPs, constitute a significant contribution. The software simply allows us to deliver these contributions in a way that maximizes community impact.


Furthermore, the community response indicates the value of our work - without any self-promotion, our public repository  (due to anonymity we don't disclose the url here) has already garnered over 120 stars on GitHub and been downloaded over 1,300 times from PyPI since its initial release approximately 4 months ago. This organic adoption highlights the research community's need for such a resource and their appreciation of our efforts in developing the first centralized benchmark for neural TPPs. We are excited to continue improving EasyTPP based on user feedback and contributing open solutions that advance the field.

---

> ### Comment · Reviewer_Xrns · 2023-11-23
> **I disagree with the other reviewers on lack of technical innovation**
>
> It seems like the main concern from other reviewers is regarding the lack of technical innovation. Considering that the paper is submitted to the primary area of datasets and benchmarks, I disagree with this concern. The amount of technical innovation is roughly what I would expect for a datasets and benchmarks paper.
>
> Furthermore, I would argue strongly for the potential significance and impact of this contribution to the growing machine learning community working on TPPs. As a researcher in this community myself, I am much more likely to make use of the contributions from this paper than, say a paper proposing a new neural network architecture for TPPs (which perhaps the other reviewers would rate more highly on technical innovation). There is a tremendous need for high quality software packages for neural TPPs, and the authors are providing just that.

---

### Author Response · Authors · 2023-11-20
**to-all message**

We thank reviewers for constructive feedback! We will kickstart our response with a few to-all messages, clarifying our contributions and innovations, presenting new discussion, as well as proposing improvements to the paper. We will address other concerns in messages to individual reviews.

Our to-all messages are organized as follows:
- [Technical Contribution and Broader Impact]: We emphasize the contribution of this work to the community.
- [Commitment]: We highlight our commitment to maintaining this library, and discuss the improvement mentioned by each reviewer.
- [Presentation Improvements]: We discuss the presentation improvements that we will execute for the camera-ready version.

---

### Meta-Review · Area_Chair_grWM · 2023-12-12

**Metareview:**

This paper reports the development of a data repository of Temporal point process.
I'm not sure if this falls under the scope of  ICLR main track. If it does then I would recommend acceptance.

**Justification For Why Not Higher Score:**

There are no technical contributions.

**Justification For Why Not Lower Score:**

It can be a  very useful resource.

---

### Decision · Program_Chairs · 2024-01-16

Accept (poster)